# Imitation of Novel Intransitive Body Actions in a Beluga Whale (*Delphinapterus leucas*): A “Do as Other Does” Study

**DOI:** 10.3390/ani13243763

**Published:** 2023-12-06

**Authors:** José Zamorano-Abramson, María Victoria Hernández-Lloreda

**Affiliations:** 1Centro de Investigación en Complejidad Social, Facultad de Gobierno, Universidad del Desarrollo, Santiago 7610615, Chile; 2Grupo UCM de Psicobiología Social, Evolutiva y Comparada, Universidad Complutense de Madrid, 28223 Madrid, Spain; vhlloreda@psi.ucm.es; 3Departamento de Psicobiología y Metodología en Ciencias del Comportamiento, Facultad de Psicología, Campus de Somosaguas, Universidad Complutense de Madrid, 28223 Madrid, Spain

**Keywords:** social learning, animal culture, production imitation, multimodal imitation, marine mammal cognition, cetaceans, beluga whale

## Abstract

**Simple Summary:**

Cetaceans, including beluga whales, are known for their unique habits and behaviors that they display within their social groups, such as group-specific tactics or vocalizations. One of the questions that has attracted the attention of researchers is whether these behaviors are learned socially, i.e., from other members of their group. In this study, we investigate the ability of a young beluga to learn and reproduce new behaviors by observing another beluga perform them. The beluga was trained to respond to the command “Do this” so that it would imitate what it had observed in another beluga whale. The results show how it was able to copy both familiar behaviors (known and previously performed) and novel behaviors (actions it had never seen or performed before) in response to the “copy” signal. This study is the first evidence of this “true imitation” (copying novel actions) ability in this species and shows that these animals can acquire new skills through this process. This ability, which is quite rare in the animal kingdom, helps us to understand how these marine mammals survive and thrive in their natural habitats and how they pass on vital information about where to live, migrate, and find food.

**Abstract:**

Cetaceans are well known for their unique behavioral habits, such as calls and tactics. The possibility that these are acquired through social learning continues to be explored. This study investigates the ability of a young beluga whale to imitate novel behaviors. Using a do-as-other-does paradigm, the subject observed the performance of a conspecific demonstrator involving familiar and novel behaviors. The subject: (1) learned a specific ‘copy’ command; (2) copied 100% of the demonstrator’s familiar behaviors and accurately reproduced two out of three novel actions; (3) achieved full matches on the first trial for a subset of familiar behaviors; and (4) demonstrated proficiency in coping with each familiar behavior as well as the two novel behaviors. This study provides the first experimental evidence of a beluga whale’s ability to imitate novel intransitive (non-object-oriented) body movements on command. These results contribute to our understanding of the remarkable ability of cetaceans, including dolphins, orcas, and now beluga whales, to engage in multimodal imitation involving sounds and movements. This ability, rarely documented in non-human animals, has significant implications for the development of survival strategies, such as the acquisition of knowledge about natal philopatry, migration routes, and traditional feeding areas, among these marine mammals.

## 1. Introduction

Cetaceans, such as dolphins and whales, are known for their advanced cognitive abilities, long lifespan, and social nature. They are capable of exhibiting group-specific behaviors such as vocal patterns and hunting tactics that are not solely determined by their individual experience with the environment or genetics. Adopting the practices of others, as opposed to relying solely on personal experience, is recognized as a crucial advantage of social interaction, fostering a form of learning, social learning, that serves as a significant catalyst in the evolution and growth of culture, e.g., [1,2,3]. In a broad sense, social learning refers to the acquisition of knowledge through observation or interaction with other individuals or their products. However, this concept encompasses various forms, each driven by different psychological processes that demand varying levels of cognitive effort. It is at this point that a unified categorization of social learning remains elusive, and there is no consensus regarding the underlying mechanisms [4,5].

One specific form of social learning is imitation, considered a cognitively challenging form of learning and a key feature of human cultural traditions [6] because it not only fosters within-group uniformity at one point in time but also allows for the accumulation of modifications over time, commonly known as the “ratchet effect” [7]. However, imitation is not a unified concept either (nor is there a definition that is fully accepted by all researchers in the field) and it can be further categorized into several types, each varying in terms of cognitive complexity [8,9,10]. In a broad sense, when an individual copies the behavior of another individual, this is considered imitation [10]. Nevertheless, the behavior being copied can be either familiar or novel, and it can be object-oriented (i.e., transitive) or body-oriented (i.e., intransitive) (and based on this, we can define various types of imitation); it is this distinction that may reflect the use of different cognitive processes during imitation [8,9,10,11]. For instance, copying a behavior that the individual already knows is less demanding than copying a completely new behavior (i.e., production learning) [7,12].

One of the gold standards for the study of animal imitation, in general, and in particular, cetaceans, is the “Do as I do” method. Originally used by Hayes and Hayes (1952) [13] to study motor imitation in a home-raised chimpanzee, this paradigm involves copying another’s action in response to a specific signal (“Do this!”). No additional cues, such as results-based information, are provided. The training process involves instructing the observer to respond to a “copy” command (“Do that”) given by the trainer. The imitation signal can be either gestural (typically involving hand gestures) or vocal. The generic “copy” sign command remains constant across all presentations or trials, regardless of the behavior demonstrated by the model that the observer is supposed to imitate [13,14]. It has been argued that for animals to successfully perform this task, they must “understand” that they are required to imitate what the model is doing. This implies that the animal subjects need to possess some level of the concept of imitation, as the method relies on the generalization of a trained signal to a conceptual instruction “copy what I am (or what the other is) doing” [15,16,17]. It is this ability that may explain the differences in success in the use of the “do as I do” paradigm among different species. Thus, efforts to train a macaque monkey to imitate on command have proven unsuccessful [18] while, conversely, this training method has yielded positive results in various other species, including great apes [14,19,20,21], dogs [22,23], and marine mammals such as dolphins [24,25,26,27], orcas [17,28,29], and even a beluga whale [30].

Most experimental studies on cetacean social learning, particularly imitation, have focused primarily on the bottlenose dolphin (*Tursiops truncatus*) [24,25,26,27]. These studies have accumulated evidence of dolphins’ exceptional imitation abilities and their proficiency to replicate both the vocal and motor behaviors demonstrated by fellow dolphins, humans, and even computer-generated stimuli. This body of evidence indicates that bottlenose dolphins are among the few non-human animal species capable of multimodal imitation, encompassing both vocal and action imitation. It has been proposed that this capacity, which transcends specific sensory inputs, represents a significant shift in neural organization, which has implications not only for imitation but also for communication as a whole [15,31].

Over the past decade, research on another delphinid species, the orca (*Orcinus orca*), has also revealed evidence of multimodal production imitation, encompassing the imitation of novel motor behaviors and vocalizations [31]. Studies have shown that orcas can imitate novel behaviors displayed by conspecifics [28] and imitate the novel vocalizations of other orcas (vocal imitative learning) or replicate human speech patterns (vocal mimicry) [29]. Additionally, they have exhibited the ability to imitate familiar intransitive actions after a delay period (up to 150 s, thus beyond what short-term memory would imply) (referred to as deferred imitation), even when interspersed with distractor (non-target) actions performed by both the demonstrator and the subjects during the retention interval [17]. The accumulating evidence suggests that orcas exhibit at least similar skill levels and, in some cases, even better performance in terms of speed in “understanding” the copy command, accuracy, and resistance to interferences compared to dolphins in terms of multimodal imitation tested under comparable conditions [17,28,29].

Another highly social cetacean that has remained relatively enigmatic in terms of its social behavior is the beluga (*Delphinapterus leucas*), the subject of the present study. These arctic cetaceans are notable for their strong gregarious tendencies, extended post-reproductive lifespans [32], and prolonged periods of maternal care [32,33]. They possess intricate vocal repertoires [34,35,36] and engage in a diverse array of interactive behaviors [37,38]. These traits collectively suggest that beluga whales inhabit complex societies that likely rely on social learning. Accordingly, it has been suggested the strong mother–calf bond may facilitate the cultural learning of natal philopatry to migration route destinations to the same summering locations after years or even decades [39]. Furthermore, beluga whale communities exhibit multi-generational groupings that encompass kin and non-kin individuals of all ages and both sexes [32,40], creating connections where diverse social learning channels can facilitate the development of cultural phenomena [41,42]. In fact, it has been suggested that this social learning dynamic could facilitate the development and perpetuation of cultural practices in this species, such as the formation of migratory circuits and the utilization of traditional feeding areas [41,42].

Supporting these suggestions, numerous investigations in more controlled conditions indicate that calf beluga whales may engage in vocal learning from their mothers [43]. These findings are reinforced by the observed cases of spontaneous cross-species vocal imitation from beluga whales to bottlenose dolphins [44] and even to human speech [45]. In a study by Ridgway et al. [45], it was documented that a beluga spontaneously replicated human sounds, prompting an examination of the physical mechanisms employed by the beluga to produce speech-like sounds. Additionally, Murayama et al. [46] conducted research in which they assessed a male beluga’s ability to mimic familiar conspecific sounds, novel artificial (computer-generated) sounds, and human speech. Their findings revealed that the study subject successfully imitated both familiar and novel sounds.

With respect to motor imitation, while some observations suggest that bubble play could be influenced by social learning [47] and that this species could possibly exhibit spontaneous yawning contagion or yawning-like behavior in response to a human [48], the first empirical evidence of motor imitation in this species came from Abramson et al. in 2017 [30]. Their study used a “Do-as-the-other-does” approach to demonstrate that beluga whales can replicate the familiar intransitive not-object-oriented body movements of conspecifics upon command [30]. This finding suggests that beluga whales, like dolphins, orcas, dogs, and apes, can be taught to replicate actions on command within a “do-as-I-do” task. However, in this initial research, it remained to be determined whether belugas share with dolphins and orcas the ability to imitate novel intransitive not-object-oriented actions, i.e., if they are capable of “productive” or “true” imitation.

The current study focuses on a follow-up of this experiment. Here, we aimed to demonstrate that belugas are also capable of imitating novel intransitive, non-object-oriented body movements on command, using a “do-as-the-other-does” paradigm. This information is a crucial factor needed to classify them as a species truly capable of multimodal imitation [31].

## 2. Materials and Methods

### 2.1. Subjects

The study was carried out on two beluga whales, Kylu and Yulka, housed at the L’Oceanografic Aquarium in Valencia, Spain. Kylu was used as the “observer” and Yulka was used as the “demonstrator”. Kylu is Yulka’s son. He was born at the L’Oceanografic Aquarium on 15 November 2016. Yulka was wild-caught in the Sea of Okhotsk, at approximately one year of age. She has been living with Kylu in the same pool since that time. The whales were fed approximately 18 kg of freshly thawed fish daily. Half of this was consumed during the experimental sessions. Both whales had participated in biological and veterinary procedures and had previously been trained through operant conditioning to perform examination and exercise behaviors, but only Yulka had participated in cognitive studies prior to this experiment, the one mentioned above on the imitation of familiar behaviors [30] and another of relative quantity judgments [49]. The whales were never deprived of food in any way.

### 2.2. General Procedure

A total of 48 sessions, with 1 to 4 sessions a day, consisting of between 3 and 12 trials with a duration of 10 to 20 min each were carried out for 474 days. The experiment was performed by two trainers, one for the demonstrator and one for the subject. The study took place in a pool with a volume of 3582 m^3^ and an area of 800 m^2^ (the same used in previous cognitive studies (see Figure 1 and, e.g., Abramson et al. [30])). The pool was equipped with a floating pontoon, measuring 2 m by 3 m, where the trainers stayed and from which they could call the subjects. This pontoon was attached to the wall of the pool and the subjects were rewarded with food or a whistle signal for correct responses. Trials were not rewarded and were repeated in case of error or partial reproductions, with a limit of four test trials of the same action in a row. The reinforcement of the demonstrator was not dependent on the subject’s response.

The study was divided into three phases: (1) training (Phase 1): The subject was trained to respond to the “copy” command given by the subject’s trainer. This was achieved using a unique hand gesture that involved touching the open palm of the left hand with the right hand; (2) testing familiar behaviors (Phase 2): The subject’s ability to generalize the “copy” command to other behaviors demonstrated by the other beluga whale (the demonstrator) was tested; and (3) testing novel behaviors (Phase 3): the ability to copy behaviors unfamiliar to the subject (behaviors that had not been exposed to him before nor had he observed them performed previously) was tested.

Table 1 shows the complete list of behaviors and their description and Table 2 shows the behaviors that were used in each of the phases and percentage of copied actions, categorized as follows: (a) Behaviors that were familiar and used during the subject’s training to respond to the trainer’s copy command (Phase 1) and during testing sessions (Phase 2) when the subject had already learned to mimic the demonstrator’s actions; (b) Behaviors that were familiar and used exclusively during testing sessions (Phases 2 and 3) and not during the initial training sessions (Phase 1); and (c) Behaviors that were novel (Phase 3), taught to the demonstrator and unfamiliar to the subject. All the familiar behaviors examined in this study were not part of the Beluga’s natural repertoire but were behaviors the subjects were trained to perform with standard operant conditioning procedures for the purpose of veterinary examinations and exercise behaviors. Novel behaviors were taught only to the demonstrator, Yulka, for the specific purpose of this study. They were trained in a separate pool from the observer, making it impossible for him to observe any part of the training of the demonstrator.

All the behaviors chosen were intransitive not-object-oriented actions. Each session was recorded on video by a camera located above or at the side of the tank, capturing the complete view of the two whale–trainer pairs and the entire aquarium area where the subjects performed their behaviors.

For Phases 1b, 2, and 3, an opaque panel was positioned between the trainers to prevent the subject from seeing the demonstrator’s trainer’s signals. The panel was checked in a pre-test to ensure that the subject could only see the demonstrator’s behavior and her own trainer’s signals. The subject’s trainer was blind to the signals presented by the demonstrator’s trainer. The chief trainer (not the testing one) and/or a researcher (both always in Phase 3) evaluated each trial and informed the trainers whether or not to reinforce the subject. Both were positioned next to the video cameras at a distance from the subject. If there was any disagreement (which occurred very few times and only in Phase 3) between the two, only the researcher’s judgment and signal on whether or not to reinforce the subject was followed.

### 2.3. Phase 1: Training the “Copy” Command

In this phase, the subject was trained to respond to the command “Do as the other does” or “copy,” given by the subject’s trainer. We followed the same protocol as used in Abramson’s et al. (2013) [28]. The subject and the demonstrator were positioned next to each other in a pool, while the two trainers faced each of them. Initially, when the demonstrator was asked to perform the behavior, the subject received both the “copy” and the behavior signals, i.e., the copy signal (initially unknown to the subject), followed by the signal of the behavior to be copied.

As the sessions progressed, the signals associated with specific behaviors were gradually spaced out until only the copy signal was used. In this phase, there was no panel, so the subject could see the signal given to the demonstrator. The training sessions began with three behaviors, GT, RO, and SO, and progressed to include BY. This phase had a total of six sessions (with 4 to 9 trials each) and a total of 31 trials.

In the seventh session (Phase 1b), the opaque panel was positioned between the trainers to prevent the subject from seeing the demonstrator’s trainer’s signals.

We used the same four familiar behaviors that were used without the panel (GT, RO, SO, and BY). In this phase, the subject received 11 training sessions (49 trials).

### 2.4. Phase 2: Generalization of the “Copy” Command to Untrained Demonstrated Familiar Behaviors

This phase consisted of 178 trials in 33 sessions. The test sessions followed the same setup as the training sessions.

The objective of the second phase was to assess the subject’s ability to apply the “copy” signal to five familiar but untrained behaviors: DA, FP, SQ, TS, and VL (as listed in Table 1). While these behaviors were already familiar to Kylu, they had never been associated with the “copy” or “Do that” signal before. In addition to these untrained behaviors, the trained behaviors from Phase 1 were also included in the testing sessions. These trained behaviors were randomly interspersed, with the restriction that no more than four trials of the same behavior occurred consecutively.

It is important to note that each session also included one to three control (“non-copy”) trials, where the subject was asked not to perform any action or to perform a different behavior from the one demonstrated. The main goal of these control trials was primarily to test whether the subject was indeed following the trainer’s signal, although they also helped to maintain the subject’s attention, prevent them from moving or looking at the other trainer’s signals, and keep them motivated.

### 2.5. Phase 3: Imitation of Novel Behaviors

This phase consisted of 35 sessions comprising 215 trials. The testing sessions followed the same setup as the Phase 2 “generalization” sessions, incorporating 1 to 4 control trials and 3 to 12 test trials. We also randomly interspersed familiar behaviors that had been used in the previous phases in order to maintain the subject’s attention and motivation in case he failed with the novel behaviors. The third phase assessed the subject’s ability to apply the “copy” signal to behaviors that were unknown to him. These behaviors were novel in the sense that the subject had not previously been exposed to them, nor had they been observed in action. This is the crucial phase in this study, as it may provide novel information regarding imitation ability in beluga whales. In essence, this phase aimed to determine whether Kylu was capable of “productive” or “true” imitation.

### 2.6. Data Coding and Analysis

Coding was performed by two experimenters. During test sessions (Phases 2 and 3), one experimenter coded the sessions in real-time while running the experiment and recorded whether the subject’s action was a correct or inaccurate match of the demonstrator’s action for each trial. In the training sessions (Phases 1a and 1b), a trainer coded the sessions in real-time while the experiment was running, and then an experimenter reviewed all the videos from each training session and checked that the subject’s actions correctly matched those of the demonstrator.

To evaluate the copy of familiar behaviors, we used a binary scale to assess the accuracy of each trial: a score of 1 indicated a full reproduction, signifying that the behavior was fully replicated, whereas a score of 0 denoted a failed reproduction, indicating that the subject executed an action that was completely unrelated to the demonstrator’s behavior.

To assess the reproduction of novel behaviors, we adopted the same framework as Abramson et al. (2013) [28]: a three-point scale, where a score of 1 indicated a complete reproduction, affirming that the behavior was faithfully replicated in its entirety; a score of 0.5 indicated a partial reproduction, where certain elements of the modeled action were missing, such as certain body parts or the orientation of the actions. Specifically, this 0.5 score was applied in the following cases: (a) for PM trials, when the subject performed a modified RO by briefly leaving the pectoral fin stationary before rolling (instead of PM, where the pectoral fin should remain motionless); (b) for LS, when the subject swam while splashing with his tail but this was executed without the sideways turning of the body; and (c) for BL, when the subject turned either underwater or during the leap—contrasting with the standard belly-down orientation during regular leaps—yet the full rotation to a belly-up position upon reaching the leaping moment it was not completely achieved. Finally, a score of 0 indicated a failed reproduction, where the subject performed an action that was totally familiar or completely unrelated to the behavior of the demonstrator.

For reliability analysis (inter-observer agreement), a second experimenter reviewed 30% of a randomly selected subset of videos from each test trial several months after the completion of the study. This second experimenter recorded whether the subject’s actions correctly matched those of the demonstrator. The inter-observer reliability was found to be exceptionally high, with a Cohen’s kappa value of 0.96 (*p* < 0.001) and an observer agreement of 0.99.

We performed exact binomial tests to determine whether the subjects successfully replicated the demonstrator’s actions above chance: (1) for each novel behavior during the generalization Phase (1b), and (2) for the entire set of behaviors combined. In the first case, the Sidak correction was applied to account for multiple comparisons.

For the analysis of familiar behaviors, we assumed that chance performance would entail successful matching on 1 divided by the total number of different familiar behaviors requested to be performed, plus the possibility of doing nothing during a trial. Thus, we assumed a chance performance rate of 1/5 for DA, the initial test behavior; 1/6 for FP, the subsequently introduced test behavior; and 1/7 for SQ, the last requested familiar behavior. When analyzing the actions together, we took the most conservative criterion, setting the chance level at 1/5. It is important to note that this criterion is rather strict, considering that the subject theoretically had the potential to perform any of the familiar behaviors he had been trained in, not just the ones requested in the test situation. Sidak adjustments were applied to account for the multiple exact binomial tests conducted, ensuring a family-wise alpha level of 0.05.

## 3. Results

### 3.1. Phase 1 Training the “Do as the Other Does” Command

Training for the “copy” signal was quite rapid, compared to our previous work in this species [30]. A total of 30 trials were performed across six sessions without the panel (Phase 1a). In Phase 1b (with the panel), all familiar behaviors (GT, RO, SO, and BY) were copied on the first trial (see Table 2 and Appendix A). All behaviors were copied above chance levels with a high degree of success: BY at 86% (*p* < 0.001, *n* = 14), GT at 92% (*p* < 0.001, *n* = 12), RO at 85% (*p* < 0.001, *n* = 13), and SO at 100% (*p* < 0.001 *n* = 10).

### 3.2. Phase 2. Generalization of “Do as the Other Does” Command

Of the five familiar behaviors introduced, three (DA, SQ, and VL) were successfully copied on the first trial (see Appendix A for DA, Appendix A for SQ, and Appendix A for VL), one (FP) on the third trial (see Appendix A), and the final one (TS) on the eleventh trial (See Appendix A). Considering all the trials in this phase, with the exception of TS that was copied at 29% (Šidák adjusted *p* = 0.26, *n* = 14), all behaviors were copied significantly above chance level: DA at 88% (*p* < 0.001, *n* = 23); FP at 48%, (*p* < 0.001, *n* = 27, SQ at 100% (*p* < 0.001, *n* = 13), and VL at 100%, (*p* < 0.001, *n* = 4), with Šidák adjustments.

Two important observations should be made about the two behaviors that were copied with less precision: FP showed irregular replication during this phase, although this was not the case in the subsequent phase. In many trials, although the subject moved in close synchrony with the demonstrator most of the time, keeping the tail fin still at his side, he did not initiate the turn, and therefore did not maintain the required position on his back of this behavior (see Appendix A).

TS exhibited the longest delay to be copied, but once it was successfully copied for the first time, it was consistently replicated in all subsequent trials (see Appendix A).

When all the trials were considered, the accuracy in matching was 83% (out of a total of 178 trials distributed across 33 sessions). When only the new familiar behaviors were considered, the percentage of successfully copied behaviors was 68% (*p* < 0.01, *n* = 84) (see Table 2).

Regarding the behaviors previously trained for replication in Phase 1, in this phase, a full match was achieved in 100% of cases for BY (*n* = 29) and RO (*n* = 16), 90% for SO (*n* = 10), and 92% for GT (*n* = 24), all *p*-values < 0.001.

Finally, in control trials (*n* = 15), the performance was 93% accurate.

### 3.3. Phase 3. Novel Behaviors

The subject correctly copied two out of three novel behaviors (75%). One behavior was completely copied on the third trial (LS) (see Appendix A), and the other was copied on the 21st trial (PM) (see Appendix A).

With regard to the latter behavior, it is important to note that we had some problems with the demonstrator’s performance from the beginning. Thus, while in the sessions in which the subject was alone, there were no problems in reproducing the PM action on cue command; in the first two experimental sessions, in the presence of the observer, the demonstrator, instead of performing the required action, PM, performed RO (i.e., instead of keeping the fin up, she rotated and ended up in a ventral position).

Therefore, from the third session onwards, we decided to ask the demonstrator to perform the new PM behavior twice on her own, i.e., before calling the subject to the platform to start the session. This modification seemed to work on some trials, but on others, including the first two after this modification, although the demonstrator started to perform the behavior correctly, and the subject paused with his pectoral fin almost doing PM and continued to do RO, the demonstrator changed the behavior to RO, thus mimicking the observer’s incorrect copying. This dynamic continued for 20 trials over 13 sessions, as Yulka continued to change the behavior to RO several times, mimicking the subject’s incorrect copy. One possible explanation for this behavior is that the demonstrator was either being influenced by the observer (who was now acting as the demonstrator) or, going a step further in speculation, she (the mother) was helping the subject (‘her son’) to obtain a reward. Indeed, the subject began to try to interact with the panel during his failed attempts, which we interpreted as a sign of his frustration.

The last behavior attempted, BL, was never fully copied after 30 trials. However, the subject jumped in close synchrony with the demonstrator on all 30 trials. In addition, on several trials (11/30), the subject’s behavior was clearly influenced by the novel behavior of the demonstrator, as he attempted to turn and move his body to the side while swimming underwater in close proximity to the demonstrator (see Appendix A).

It is important to note that for a copy to be considered a complete match we adopted a fairly strict criterion: the subject had to reproduce exactly the same chain of motor actions, as well as the position and location in the pool where the actions were performed. However, a closer examination of the first attempts revealed that, even on the trials where their responses were incorrect, almost all of the subjects’ actions appeared to be influenced by the demonstrator in some way, i.e., there was always some motor component that corresponded to the demonstrated behavior (as shown in Appendix A).

## 4. Discussion

This study provides the first experimental evidence of productive action imitation in the beluga whale. It yielded a positive result in a crucial test scenario, an element missing in our previous research on this cetacean species [30]. In the current research, a young beluga whale, Kylu, served as the observer and demonstrated not only the ability to swiftly learn the “Do as the other does” command to replicate familiar actions but also the ability to mimic unfamiliar actions observed performed by the demonstrator. Specifically, Kylu began responding to the trainer’s command after an average of 20 trials (ranging from 11 to 37) and successfully replicated the behavior of a conspecific, both familiar and novel, no later than the 21st attempt.

While previous “Do as I Do” studies have demonstrated motor imitation success, there is considerable variability in the rate at which individuals acquire the ability to replicate commands across different studies. For example, Kylu achieved success relatively quickly, in contrast to Yulka, the other beluga subject trained with this command [30]. Similarly, compared to dolphins, Kylu’s success was also achieved relatively quickly. In a study by Bauer and Johnson [24], it was noted that the two dolphins involved in their research required “hundreds of trials” and “more than 1000 trials”, respectively, to grasp the mimic command. Moreover, Kylu’s performance in learning the copy command is not so far from that achieved in orcas, reported by Abramson et al. [28], in which three orcas started copying the demonstrator’s actions from the very beginning.

One possible explanation for the challenges some species face in acquiring this command is that success in this task depends on the acquisition of conceptual learning, which forms the foundation for generalizing the trained “copy what I am (or what the other is) doing” signal to different behaviors [15,16,17]. Remarkably, the subject had no difficulty in understanding this command, even at his young age. This accelerated learning rate could be attributed to Kylu’s youth and his innate tendency to mimic his mother [43,48,50,51]. Furthermore, this predisposition may have been further reinforced when Kairo, the 55-year-old male and the only other whale beside the mother–calf pair in the pool, unfortunately died on 26 February 2022. This situation left Kylu alone with his mother, Yulka, who was now the only remaining whale in the pool for social interaction and learning.

Similar to Yulka’s performance, Kylu showed remarkable consistency in matching accuracy after producing his first correct copy, a pattern similar to that observed in dolphins by Bauer and Johnson [24] (81% and 84%) and orcas by Abramson et al. [28] (83%, 81%, and 94%), as shown in Table 2. Once Kylu mastered the copy command in this study, he demonstrated the ability to extend it to other familiar actions. Impressively, he successfully imitated five other familiar, but untrained behaviors performed by the conspecific demonstrator, achieving a perfect 100% success rate. This rapid and sustained high level of correct performance continued after his first successful match, as documented in Table 2.

Compared to Yulka, Kylu performed significantly better, achieving full matches above the chance level for a higher percentage of demonstrated behaviors. Specifically, Kylu copied 100% of demonstrated behaviors above chance level, while Yulka only managed to do so for 64% of them. When analyzing each familiar test behavior individually, Kylu also outperformed Yulka, copying four out of five familiar behaviors on the first trial (DA, FP, VL, and SQ), as shown in Table 2 and Appendix A. The TS behavior took Kylu until the 11th trial to imitate (see Table 2 and Appendix A). In contrast, Yulka’s fastest imitation occurred in the 12th trial, while the slowest occurred in the 35th trial.

Compared to dolphins, Kylu’s continued high level of correct performance after the initial successful match (as shown in Table 2) was very similar to the results of Jaakkola et al.’s [25] study of motor imitation in dolphins. In their “sighted” condition, which was similar to our familiar behavior condition, dolphins achieved a 61% matching accuracy across 19 motor behaviors. Compared to orcas, Kylu’s generalization of the copy command closely resembled that of orcas in Abramson et al.’s [28] study. In this study, familiar behaviors performed by the demonstrator were copied before the 8th trial, with many of these being copied on the first attempt (ranging from 57% to 93%). More remarkably, Kylu’s imitation of novel actions was also quite close to the orcas' performance, as Kylu successfully copied 75% of the demonstrator’s novel actions tested (*n* = 2/3), consistently performing above the chance level for two out of three novel behaviors tested.

The results of this study are also consistent with and build upon those observed in dolphins and orcas when imitating untrained novel actions in a similar “do-as-other-does” task. Notably, Kylu was able to copy a novel behavior on the third trial (LS), similar to a dolphin in Xitco’s 1988 [27] experiment, where one subject mimicked one out of three novel behaviors on the third trial, as reported by Herman in 2002 [15]. In contrast, two dolphins in Bauer and Johnson’s 1994 [24] study failed to imitate novel behaviors, whereas the beluga whale in this study successfully copied two out of the three untrained actions presented to him.

In terms of familiar actions, Kylu had difficulty copying one out of the five actions. Interestingly, we cannot attribute this challenge to the complexity of the behavior, as others such as BY, which were equally complex, posed no issues for him. We speculate that the difficulty with a Tail Splash (TS) may have arisen because the model performed this behavior rather shyly and with a gentle fluke splash, which Kylu may not have noticed. Notably, when we conducted separate training sessions with the model to increase the intensity of the splash, Kylu began to replicate TS correctly.

Similarly, although he had more difficulty, Kylu eventually demonstrated proficiency in fully matching both simple novel actions (e.g., PM Appendix A) and more complex novel actions (e.g., LS Appendix A). However, while he copied much of it, he never fully succeeded in reproducing the more complex action presented to him (BL). On the other hand, the initial part of Pec Mimic (PM) was very similar to the behavior of Rolling Over (RO). Paradoxically, this similarity may have made copying more challenging for Kylu, possibly causing him to fixate on the initial phase of the demonstration. It is even possible that the model sometimes switched to rolling over at the beginning to make it easier to elicit the response or to help Kylu obtain a reward. Despite some errors on several trials, Kylu tended to reproduce the primary components of the demonstration (Table 2).

Among the novel behaviors presented, the Back Leap (BL) was by far the most complex. Kylu was asked to dive to the bottom of the pool and then turn 180° and leap backward out of the water. It is important to note that in all trials Kyku performed a jump almost together with Yulka, often quite synchronously, which is consistent with what has been proposed about the central role of behavioral synchrony in the evolution and development of cetacean imitation [31,52].

In addition, in 11 of the 30 trials, Kylu tried to turn either at the beginning of the dive, or even underwater when reaching the bottom of the pool, several times, even diving and swimming turned sideways next to Yulka for a few seconds (see Appendix A). However, even when he did this, he always returned to his normal position when he came and jumped out of the water, so he never managed to turn completely when he came to the surface and therefore jumped and landed on his back. This may have been due to the complexity of the action and the uncertainty and fear that could have caused him to turn and jump in this way, and he could easily have fallen backwards onto his mother given how close and synchronous Kylu performed this maneuver. In fact, when it comes to artistic gymnastics, it is quite common in humans for athletes to struggle with the emotion of fear [53]. Indeed, after several failures by her son, Yulka changed the inverted jump to the normal one several times. As in the previous case, we do not know whether this was due to mere contagion with Kylu, which is most likely, or whether she did it deliberately to help him. In addition, it should be noted that in both human gymnastics and in Yulka, the training of an inverted jump is achieved by the positive reinforcement of small successive approaches. In Yulka’s case, this was achieved after a training process that lasted about six months, with 7 sessions per week, with 4 to 6 trials per session (42 to 50 sessions, 168 to 300 trials in total). Furthermore, the training process consisted of eight steps in all, but in the last two of which, a target was used to shape Yulka’s jump. Yulka even had the help of a second trainer with a target in another part of the pool to guide her and eventually get her to jump backward.

It is crucial to emphasize that in delineating novelty, there will invariably exist a certain level of resemblance to the actions previously undertaken by the observer [54]. Furthermore, the replication of novel actions remains a viable possibility through the identical learning mechanisms employed for familiar actions [55,56]. In our study, most of all the motor actions, whether they were familiar or novel, shared a common characteristic: they were primarily intransitive in nature actions, i.e., they were body-oriented and did not involve external objects [9,30]. It has been proposed that the challenge of imitating intransitive actions arises primarily from the fact that the observers lack many environmental cues that could help align their perceptual representation of the action with that of the demonstrator [57,58,59,60]. Previous experimental studies in animal imitation, involving apes [14,19] and dogs [22,23,59], have also shown that familiar intransitive actions tend to be more difficult to imitate than familiar transitive actions.

In the current investigation, the beluga whale observed the demonstrator’s bodily actions and, in response to the “Do what the other is doing” command, either made a choice from its existing repertoire of untrained familiar actions or effectively extended the copy command to include “novel” transfer behaviors, i.e., for production imitation, showing a greater degree of evidence for its ability to generalize this copy command to novel intransitive actions.

Furthermore, these findings suggest that beluga whales, like dolphins and orcas, are among the select group of mammalian species, along with humans, known to exhibit “productive” cognitive skills in vocal and motor imitation. This capacity for “multimodal imitation”, encompassing the replication of novel kinesthetic motor actions and vocalizations, is a rarity in the non-human animal world. It serves as an important marker in the evolution of human communication, as both vocal learning and visual–gestural imitation played pivotal roles in the development of human speech and singing. As cetaceans are among the few mammals known to possess the multimodal imitation skills discussed, they are proving to be a valuable model for shedding light on one of the fundamental unanswered questions in human evolution: the development of speech and singing. Their abilities provide insights into the evolutionary steps necessary for the emergence of human forms of linguistic and musical communication.

## 5. Conclusions

This study revealed a beluga whale’s ability to imitate novel intransitive non-object-oriented body movements performed by a conspecific, i.e., the ability for productive imitation. This finding builds upon our prior research on the imitation of familiar actions within this species, taking our understanding to a new level. It also extends previous findings in beluga whales [30,46] and the existing body of knowledge on motor and vocal imitation in other cetacean species, such as bottlenose dolphins [30,46] and orcas [17,28,29]. Furthermore, it underscores the exceptional status of beluga whales, dolphins, and orcas, as they join humans in the select group of mammals renowned for their multimodal imitative learning capabilities, encompassing both productive vocal and motor imitation skills. Finally, this study complements field research by shedding light on the significance of this multimodal imitative learning capacity as a potential mechanism contributing to the acquisition, behavioral diversity, and transient stability observed in field populations of belugas and other cetacean group cultures [39,42,60].

## Figures and Tables

**Figure 1 animals-13-03763-f001:**
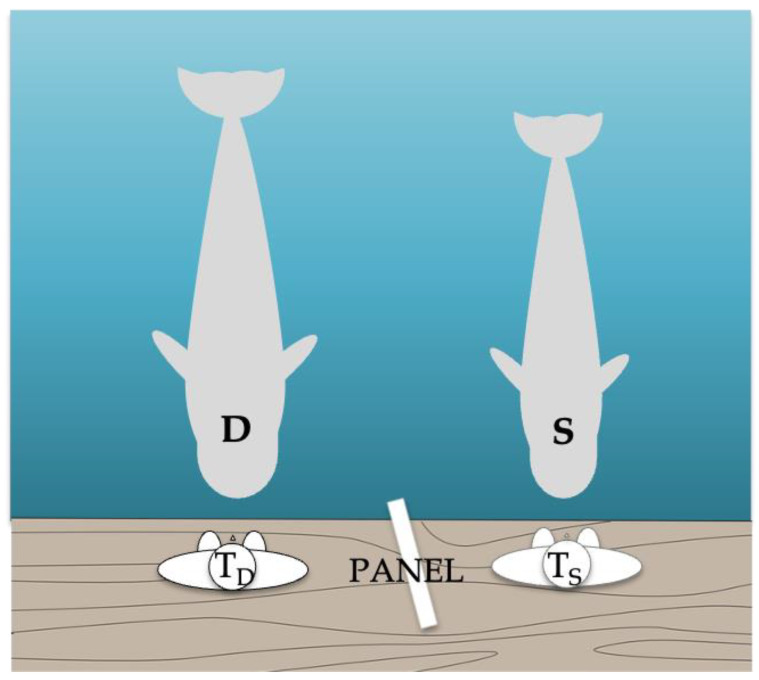
Experimental setup. Two trainers (TD and TS; D for demonstrator and S for subject) were positioned on different sides of an opaque panel 2 m long × 91 cm high placed in a position in which S and D could see each other and their own trainer but could not see the other trainer’s commands.

**Table 1 animals-13-03763-t001:** Behaviors tested in each phase.

	Description
Training Phase 1	
Roll Over (RO)	Turn over, ventral side up, horizontally (parallel to the water surface), and hold this position
Greeting Tail (GT)	Dive downward to a vertical position with the tail fluke protruding from the water and shake it
Song (SO)	Emit a whistling sound (vocalize out of the water)
Bye-Bye (BY)	Turn to one side, raise one pectoral fin out of the water, and wave it back and forth from the surface while swimming away from the starting point
Testing Phase 2	
Dance (DA)	Rise vertically on water, half of the body on the surface, and roll continuously in 360°
Fluke Present (FP)	Roll 180° to ventral up position, turn 180° to head tail position, and take the tail out of the water next to the trainer position
Tail Splash (TS)	Slap tail continuously on the water's surface
Squirt (SQ)	Split water out above the water’s surface
Ventral Leap (VL)	Swim to the left, dive to the bottom of the pool, and then jump out of the water
Testing Phase 3	
Pec Mimic (PM)	Raise pectoral fin out of the water while holding it still for a few seconds
Back Leap (BL)	Swim to the right and dive to the bottom of the pool and then turn 180° and jump backwards out of the water
Lateral Splash (LS)	Turn to the left, parallel to the platform, and swim away from the starting point, constantly slapping the surface with the tail fluke and flapping the right pectoral out of the water

Every behavior is described as the starting point of the animal facing the trainer while lying horizontally on the water’s surface and in a perpendicular position to the pool wall.

**Table 2 animals-13-03763-t002:** Total number of trials.

	No. of Trials	% Copied Actions	Trial No. of 1st Full Copy
Training (Phase 1b)			
BY	14	86	1
GT	12	92	1
RO	13	85	1
SO	10	100	1
Phase 2 Testing			
DA	26	88	1
FP	27	48	3
TS	14	29	11
SQ	13	100	1
VL	4	100	1
Behaviors used in Phase 1			
BY	29	100	
GT	24	92	
RO	16	100	
SO	10	90	
CO	15	93	
Phase 3			
Familiar Behaviors used in Phases 1 and 2			
BY	6	100	
GT	13	100	
RO	22	100	
SO	15	100	
DA	7	100	
FP	14	93	
CO	33	100	
Novel Behaviors		% Partial copied	First full copy
LS	18	66 (12)	3
BL	30	38 (11)	
PM	30	20 (6)	21

For Training and Phase b: Number of trials and % copied actions. For Phase 3: Number of trials, % of partial copies, and first trial fully copied. Abbreviations: roll over (RO), greeting tail (GT), song (SO), bye-bye (BY), dance (DA), fluke present (FP), tail splash (TS), squirt (SQ), ventral leap (VL), pec mimic (PM), back leap (BL), lateral splash (LS).

## Data Availability

The data presented in this study are openly available in FigShare at https://doi.org/10.6084/m9.figshare.24418315.v1.

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
