# Peer review of "Imitation of Novel Intransitive Body Actions in a Beluga Whale (Delphinapterus leucas): A “Do as Other Does” Study"

_animals, 2023, doi:10.3390/ani13243763_

Round 1
Reviewer 1 Report
Comments and Suggestions for Authors
It was a pleasure to review this manuscript. It provides an excellent report of a timely study. It is a nice follow up on the previous report regarding imitation of object-focused behaviors that was published in PLOS. I do, however, offer the following points for consideration.
Line 157: The authors state that Yulka has been in captivity since 1 year of age, but they also state that she was “raised by her mother in the .. sea.” These two statements are incompatible. Surely these authors know nothing about that whale’s rearing during its first year in the wild, and evidently her mother was not with her during most of her rearing. May I suggest the following: Yulka was wild-caught in the Sea of Okhotsk, at approximately one year of age.
Line 166: “Forty-eight sessions…” Please also include the number of days of training, how many sessions per day, and the number of days between training days.
Line 198: “Novel behaviors were taught only to the demonstrator…” This claim is of importance to this study. Therefore, more detail needs to be provided on how this was accomplished. Was Kylu in the same pool while Yulka was trained? How can you assert that it was impossible for him to observe any portion of her training?
Line 194: Phase 3 is described as providing novel behaviors. However, both Table 2 and Line 250 make it clear that familiar behaviors were also included in Phase 3. This should be clarified.
Table 2: The column heading “First Full Copy” is vague and difficult to interpret. May I suggest the following: Trial Number of First Full Copy.”
Lines 223-229: State whether Kylu’s trainer was “blind” to the signals presented by Yulka’s trainer. And describe the position of the “chief trainer” and the “researcher.” Since these two individuals were not blind to the Yulka’s signals, their position relative to Kylu is important. Lastly, since two individual (chief trainer and researcher) judged Kylu’s behaviors, what process occurred if they disagreed?
Line 232: “176 trials in 33 sessions” is specified. Yet Line 238 has “178 testing trials … across 33 sessions.” Please clarify.
Line 509: Here the authors describe this study as pertinent to “multimodal” imitation. Yet, the authors provide no evidence that more than one sensory modality was used by the whales. Indeed, none of the three novel behaviors required anything other than vision, and only one of the previously trained behavior specifically involved sound. The multimodal claim is unwarranted, and it is unnecessary to justify the value of this report.
In addition to those issues, there were a number of typographical errors. In an effort to be helpful, I offer the following:
Line 58: “foster” should be “fosters”
Line 60: there is an open parenthesis that does not close
Line 137: “specie” is not a word. It should be corrected to “species”
Line 154: “haused” should be “housed”
Line 157: “in” should be “on”
Line 196: “behavior” should be “behaviors”
Line 237: The passage “with the "copy" or "o that" signal before.” is scrambled and should be corrected.
Line 343: “one” should be “own”
All of these issues notwithstanding, these authors should be congratulated on completing this study, and for preparing this manuscript. Once these issues are addressed, it will be a pleasure to endorse it.
Comments on the Quality of English Language
In addition to those issues, there were a number of typographical errors. In an effort to be helpful, I offer the following:
Line 58: “foster” should be “fosters”
Line 60: there is an open parenthesis that does not close
Line 137: “specie” is not a word. It should be corrected to “species”
Line 154: “haused” should be “housed”
Line 157: “in” should be “on”
Line 196: “behavior” should be “behaviors”
Line 237: The passage “with the "copy" or "o that" signal before.” is scrambled and should be corrected.
Line 343: “one” should be “own”
Reviewer 2 Report
Comments and Suggestions for Authors
One recent relevant citation on imitation learning in cetaceans that should be cited is:
Jones, L.S., Stephenson, T.A., Zoidis, A.M. and Todd, S.K., 2022. Drone Observations of a Mother–Calf Humpback Whale (Megaptera novaeangliae) Pair Synchronous Feeding in the Bay of Fundy, Canada. Aquatic Mammals, 48(6), pp.716-719.
Remove the word "tender" on line 391. Use an adjective that is less flowery.
Reviewer 3 Report
Comments and Suggestions for Authors
This original article was submitted to the special issue of Advances in Marine Mammal Cognition and Cognitive Welfare which invites articles on cognition such as the one under review.
In this study, the authors trained a beluga whale to copy the behaviors demonstrated by another beluga, its mother. It was speculated that social learning could play a role in belugas making them ideal subjects to study imitation. As a first study on imitation in belugas (Abramson et al. 2017) had not tested the beluga’s ability to copy entirely novel behaviors, the authors set out to test exactly this aspect in the current study. The beluga was first trained on the copy command, it was then asked to generalize this command to additional, but familiar behaviors. In the last phase of the experiment, the beluga was asked to copy three novel behaviors: he copied two behaviors on trial 3, and 21, but failed to copy a third behavior even after 30 trials. The authors conclude that belugas are able to imitate novel intransitive body movements on command.
While I think that the study is generally addressing an interesting topic, my major worry is central: For me, the authors should clearly define when they consider the subject to have imitated successfully/have acquired an imitation concept. In my opinion, for concept formation, it is crucial to look at the very first test trial in which the subject was asked to copy a novel behavior and here it should score with 1 – in my opinion, it is not a matter of statistics. If I am wrong, please explain as a response to my comment.
Introduction
66 Could you please add a reference where these types of imitation are explained?
Material and methods
163 Please mention the studies the beluga has previously been part of to know whether these were closely linked to imitation or totally different.
174 As you have introduced “TD” for “demonstrator, please use this abbreviation consistently in the text; also holds for “TS” for “subject”. It would make the text easier to read if you also introduced for example “TTS” for the “trainer of TS” and “TTD” for the “trainer of TD”.
176 I would prefer to see a photo of the actual setup and the belugas in it. Maybe a still image of the videos could be used.
180ff Please also explain the training for TD (maybe at position 199, more details at position 253 – what does it mean that “they had not been observed in action”?). When and where was TD taught the behaviors that were then classified as novel for TS? Was there any possibility for TS to watch this training of TD? Or is it possible to separate the animals for such a training? Can you exclude that TD showed the behaviors outside the training sessions and that the behaviors were “culturally transmitted” from mother to son? (In the introduction, you reason that this cultural transmission might indeed happen in belugas.)
203 In Tab.2, under Phase 3, shouldn’t it read “familiar, used in Phase 2” or “familiar, used in Phase 1 and 2”? What do the numbers behind the percentages under “% partial copy” indicate?
210ff How did the trainers proceed when TS did not copy the behavior? Moreover, when the beluga did not copy what did it do? Did it partly show the behavior or did it show entirely different behaviors? These latter questions are important in the context of the results of phase 3. Additionally, was the trainer of TS aware of the behavior asked from TD before the trainer of TS saw the behavior of the demonstrator? At some point, the trainer of TS saw TD performing the behavior – please discuss whether it would have been crucial or not to exclude the trainer of TS to observe what TD is doing – these details are important with respect to secondary cue giving.
224 This control is crucial. Please describe in more detail how the control was performed, for which period of time, how did the beluga respond etc. Would it have been impossible to mask the TS during the time the command was given to the TD and only allow TS to watch TD while it performed the behavior?
248 Why did the authors decide to include the test trials in control trials and not in copy trials? Usually the number of test trials versus “other” trials is small in testing sessions, why did the authors perform that many test trials?
271 Were the behaviors asked from the belugas distinct from each other or “kind of overlapping” (i.e. if one behavior was performed with a score of 0.5, this was comparable to a different behavior performed with a score of 0.5?)? Most likely this was (although I read 275 differently) considered but I would like the authors to mention this explicitly, maybe in the context of Tab.1 or here – or to mention which behaviors were distinctly different and which overlapped. How many behaviors can the belugas perform on command? This would give the reader an impression if the list of familiar behaviors is only a small subset or all the behaviors the beluga can perform on command.
Results
301 Should the percentages be the same as in Tab.2? In Tab.2, I cannot read 86%, 92%, 85% and 100%. Please check text and table carefully.
332 If the beluga copied the behavior on trial 3 or 11, what did the beluga do in the proceeding trials? Did it partially copy or did it show an entirely different behavior? Please mention this in the results section.
361 Is it possible to perform the behavior in the exact same position? I am asking as the belugas performed the actions at the same time – and this leads me to a crucial question: would it have been better to wait for TD to complete the entire behavior before asking TS to copy? Could this procedure have eliminated some of the problems you describe in the results sections for some behaviors?
365 The formulation “appeared to be influenced in some way by the demonstrator” is very vague, please specify.
Discussion
443 From the videos, it is clear that the copy command was not given AFTER TD had performed the behavior but BEFORE the completion. With respect to the mentioned similarity between some behaviors, please discuss whether waiting to copy a behavior after the completion of TD’s behavior might have solved the problem. Can you exclude any type of “communication” between the belugas during the experiment?
452 If they performed the jump together, how can this be imitation?
Comments on the Quality of English Language58 Please correct to “fosters”.
60 Please close the brackets at the correct position in the sentence.
91 Please delete “in”.
122 Please delete the period before the references.
125 Please insert a space before the references.
137 Please correct to “species”.
169 Please correct to “m³”.
237 Please correct to “do that” (?).
305 Please correct to “copied”.
361 Please either use “perform” or “reproduce”.
428 Please correct consistently to “[]” for references in this paragraph.
502 Please correct to “beluga whale’s ability”.
